

# The invasive red-eared slider turtle is more successful than the native Chinese three-keeled pond turtle: evidence from the gut microbiota

Yan-Fu Qu[1], Yan-Qing Wu[2], Yu-Tian Zhao[1], Long-Hui Lin[3], Yu Du[1,4], Peng Li[1], Hong Li[1] and Xiang Ji[1]

[1] Jiangsu Key Laboratory for Biodiversity and Biotechnology, College of Life Sciences, Nanjing Normal University, Nanjing, Jiangsu, China
[2] National Key Laboratory of Environmental Protection and Biosafety, Nanjing Institute of Environmental Sciences, Ministry of Ecology and Environment, Nanjing, Jiangsu, China
[3] Hangzhou Key Laboratory for Ecosystem Protection and Restoration, College of Life and Environmental Sciences, Hangzhou Normal University, Hangzhou, Zhejiang, China
[4] Hainan Key Laboratory of Herpetological Research, College of Fisheries and Life Science, Hainan Tropical Ocean University, Sanya, Hainan, China

Corresponding author
Xiang Ji, jixiang@njnu.edu.cn

## ABSTRACT

**Background**. The mutualistic symbiosis between the gut microbial communities (microbiota) and their host animals has attracted much attention. Many factors potentially affect the gut microbiota, which also varies among host animals. The native Chinese three-keeled pond turtle (*Chinemys reevesii*) and the invasive red-eared slider turtle (*Trachemys scripta elegans*) are two common farm-raised species in China, with the latter generally considered a more successful species. However, supporting evidence from the gut microbiota has yet to be collected.

**Methods**. We collected feces samples from these two turtle species raised in a farm under identical conditions, and analyzed the composition and relative abundance of the gut microbes using bacterial 16S rRNA sequencing on the Roach/454 platform.

**Results**. The gut microbiota was mainly composed of Bacteroidetes and Firmicutes at the phylum level, and Porphyromonadaceae, Bacteroidaceae and Lachnospiraceae at the family level in both species. The relative abundance of the microbes and gene functions in the gut microbiota differed between the two species, whereas alpha or beta diversity did not. Microbes of the families Bacteroidaceae, Clostridiaceae and Lachnospiraceae were comparatively more abundant in *C. reevesii*, whereas those of the families Porphyromonadaceae and Fusobacteriaceae were comparatively more abundant in *T. s. elegans*. In both species the gut microbiota had functional roles in enhancing metabolism, genetic information processing and environmental information processing according to the Kyoto Encyclopedia of Genes and Genomes database. The potential to gain mass is greater in *T. s. elegans* than in *C. reevesii*, as revealed by the fact that the Firmicutes/Bacteroidetes ratio was lower in the former species. The percentage of human disease-related functional genes was lower in *T. s. elegans* than in *C. reevesii*, presumably suggesting an enhanced potential to colonize new habitats in the former species.

# INTRODUCTION

The past few years have witnessed much attention paid to the gut microbiota of aquatic animals including invertebrates (*Meziti, Mente & Kormas, 2012*; *Wang et al., 2019*; *Gao et al., 2020*), fish (*Divya et al., 2012*; *Xing et al., 2013*; *Egerton et al., 2018*), reptiles (*Ahasan et al., 2018*; *Zhang et al., 2019*; *Scheelings et al., 2020*) and mammals (*Nelson et al., 2013*; *Delport et al., 2016*). The gut microbes have a mutually-beneficial relationship with their hosts. Numerous studies have confirmed that the gut microbes encode 10 to 100-fold more distinct genes than their host genome (*Turnbaugh et al., 2006*; *Qin et al., 2010*), and that they are involved in a large number of host biological processes related to nutrient absorption (*Kartzinel et al., 2019*), gut homeostasis maintenance (*Buchon, Broderick & Lemaitre, 2013*), growth (*Videvall et al., 2019*) and even behavioral expression (*Ye et al., 2014*). Therefore, it is of great significance to study the coevolutionary relationship between the gut microbes and their hosts.

The community structure and relative abundance of gut microbes vary among hosts. In aquatic animals, for example, the dominant phyla are Proteobacteria, Bacteroidetes, Tenericutes and Firmicutes in crabs (*Hong et al., 2020*; *Wei et al., 2019*; *Wei et al., 2020*), Bacteroidetes, Firmicutes and Proteobacteria in shrimps (*Rungrassamee et al., 2014*), fish (*Egerton et al., 2018*), sea turtles (*Campos et al., 2018*; *McDermid et al., 2020*) and sea lions (*Delport et al., 2016*), and Proteobacteria and Firmicutes in sharks (*Givens et al., 2015*). Each gut microbial phylum may have unique functional roles. For example, members of the phylum Proteobacteria contribute to the breakdown and ferment of the complex sugars and are related to the biosynthesis of vitamins for their hosts (*Colston & Jackson, 2016*), microbes of the phylum Bacteroidetes improve the digestive efficiency in both herbivorous and carnivorous species by degrading the complex macromolecular matter (*Colston & Jackson, 2016*), members of the phylum Firmicutes contribute to the production of enzymes involved in fermenting vegetative material and have the potential to fabricate vitamin B (*Rowland et al., 2018*), and microbes of the phylum Tenericutes are involved in the nutrient processing of their hosts (*Colston & Jackson, 2016*). Taken together, gut microbes play a vital role in maintaining the normal life of the host.

Gut microbes are affected by many factors, including the host's genetic background, food habit (*Kohl et al., 2017*), ontogenetic stage (*Videvall et al., 2019*), gender (*Mueller et al., 2006*) and health status (*Lin et al., 2019*), and vary seasonally (*Kohl et al., 2017*; *Kartzinel et al., 2019*). The phylogenetic dependence of the gut microbial communities has been documented in fish (*Givens et al., 2015*), reptiles (*Hong et al., 2011*), birds (*Capunitan et al., 2020*) and mammals (*Kartzinel et al., 2019*). In mammals, for example, the gut microbial diversity is higher in herbivorous species than in carnivorous species (*Kartzinel et al., 2019*). In humans, microbes of the phylum Firmicutes are positively correlated with latitude and obesity, whereas members of the phylum Bacteroidetes show opposite correlations

(*Ley et al., 2005*; *Suzuki & Worobey, 2014*). Moreover, the gut microbial profiles at the different life stages are complicated by the natural dynamics and complex interactions between intrinsic and extrinsic factors. For example, early growth, household location, and antibiotic experiences during pregnancy are correlated with the early gut microbial composition in humans (*Vatanen et al., 2019*). Seasonal variation in diets may lead to changes in the gut microbiota (*Kartzinel et al., 2019*), and exposure of host animals to artificial environments (e.g., animals in captivity) may lead to an increase in the abundance of human disease-related functional genes in the gut microbes (*Tang et al., 2020*; *Zhou et al., 2020*). Taken together, the gut microbial community is affected by the host's genetic background, diets and individual status and, in turn, affects the physiological, behavioral and even evolutionary processes of the host.

In the present study, we compared gut microbes between the native Chinese three-keeled pond turtle (*Chinemys reevesii*) and the invasive red-eared slider turtle (*Trachemys scripta elegans*) using bacterial 16S rRNA sequencing on the Roach/454 platform. Both are well studied species and among common turtle species farm-raised in China as food, pets and traditional medicine, with *T. s. elegans* well known as a very successful invasive species in many places around the world (*Cadi et al., 2004*). The red-eared slider turtle is believed to be introduced to mainland China through Hong Kong as a pet and source of food in the 1980s and has become established in many parts of China (*Ma & Shi, 2017*). The turtle is generally considered to be more successful than native freshwater turtles such as the Chinese three-keeled pond turtle and has be listed by the World Environmental Protection Organization as one of the 100 most harmful invasive species (*Ma & Shi, 2017*). However, supporting evidence from the gut microbiota has yet to be collected.

## MATERIALS AND METHODS

### Sample collection

This study was performed in accordance with the current laws on animal welfare and research in China, and was approved by the Animal Research Ethics Committee of Nanjing Normal University (IACUC-200422).

We obtained three juvenile *C. reevesii* and three juvenile *T. s. elegans* in March 2013 from a turtle farm in Hangzhou, Zhejiang, East China, and brought them to our laboratory in Nanjing, where they were individually housed in $340 \times 230 \times 200$ mm (length $\times$ width $\times$ height) aquariums placed in a room for six months. Water temperatures varied from $26 - 30\ ^\circ C$ (with a mean of $28\ ^\circ C$), and photoperiod inside the room was on a cycle of 12 h light and 12 h dark. Turtles were fed with commercially sold food (10% water, 60% proteins, 5% lipids, 5% carbohydrates and 20% minerals; (*Li et al., 2013*) at an average ration of 1.5% body mass daily. At the end of the experiment, body masses varied from 56–70 g (with a mean of 64 g) in *C. reevesii*, and from 118–136 g (with a mean of 125 g) in *T. s. elegans*. We collected fecal samples from each turtle on 25th September 2013 and stored them at $-80\ ^\circ C$ until later DNA extraction.

## Sample sequencing

Genomic DNA was extracted using the Qiagen TM QIAamp DNA Stool Mini Kit (Hilden, Germany) following the manufacturer's instructions. The V1–V3 region of the 16S rRNA gene was chosen for the amplification and subsequent pyrosequencing of the PCR products. The following 16S rRNA primers were used for the PCR reaction: 8F (5′-AGAGTTTGATCCTGGCTCAG - 3′) and 533R (5′- TTACCGCGGCTGCTGGCAC- 3′) for the V1-V3 regions. Each primer included "barcode" sequences to facilitate the sequencing of products in the Roche/454 GS FLX+ system (454 Life Sciences, USA). The fusion primer sequences were 5′-454adapter-mid- CCTACGGGAGGCAGCAG-3′(forward) and 5′-454adapter-mid- CCGTCAATTCMTTTRAGT-3′(reverse). DNA (20 ng) from each sample was used for amplification in 25 µl reactions that contained 2.5 µl 10-fold reaction buffer, 40 ng of fecal DNA, 10 µM each primer, 0.625 U Pyrobest polymerase (Takara), and 5000 µM concentration of each of four deoxynucleoside triphosphates. PCR reactions were started by an initial denaturation at 94 °C for 5 min followed by 27 amplification cycles (94 °C for 30 s, 45 s at annealing temperature, 72 °C for 1 min) and a final extension step for 7 min at 72 °C. Subsequently, PCR products were examined for size and yield using agarose gel in TAE buffer (20 mM Tris–HCl, 10 mM sodium acetate, 0.5 mM $Na_2$EDTA, pH 8.0). Quantification of the PCR products was performed by using the PicoGreen dsDNA BR assay kit as recommended by the manufacturer. Then, the V1–V3 region of 16S rRNA was sequenced on a Roche GS-FLX 454 platform (Roche, Shanghai, China) according to 454 protocols.

## Quality control and data standardization

The pyrosequencing data were optimized by control standards according to the following criteria: (1) the raw reads must perfectly match the primers we used; (2) the barcode sequences with quality average were at least 25%; (3) the range of read length was between 320 to 800 bp nucleotides except for barcodes and primers; (5) consecutively identical bases did not exceed six and excluding undetermined bases. A total of 52,469 high-quality reads with an average length of 480 bp nucleotides were obtained. These sequences were then submitted to the National Center for Biotechnology Information (NCBI) Bioproject database (SRA accession number PRJNA645767).

We used Usearch 7.0 to conduct analysis of the operational taxonomic units (OTUs) (*Edgar, 2010*). We first extracted nonrepeative sequences from high-quality data to reduce redundant calculations in the analysis and removed the singletons. Then, these available sequences were clustered into OTUs according to the similarity criteria of 97%, and chimeras were removed in the clustering process to obtain the representative sequences of OTUs. Finally, map all available sequences for each sample to representative sequences of OTUs to obtain OTU tables for further analysis. Moreover, RDP Classifier 2.2 (*Quast et al., 2012*) performed a taxonomic analysis of the OTU representative sequences against the Greengenes 135/16S Database at the confidence threshold of 70% to obtain the taxonomic information corresponding to each OTU. To avoid large partial sample deviations, we retained OTUs with the number of OTU greater than 5 in at least 2 samples for further analysis

The alpha diversity index under different random sampling was calculated by using mothur 1.30.2 (*Schloss et al., 2009*) and visualized using the R 4.0.1 (*R Development Core Team, 2020*) to assess the adequacy of sequencing data. We standardize the OTUs abundance information according to the sample with the least sequence number for further analysis.

## Alpha and beta diversity estimation

The alpha-diversity indexes including the community richness (ACE index), diversity (Shannon diversity), evenness (Shannoneven index), and coverage (the Good's coverage index) were calculated using mothur software for each sample. These estimators were presented visually in the form of pictures using the R software platform. Bartlett test was used to test whether the data is of equal variance and the equal-variance $t$-test or the heteroscedasticity $t$-test was conducted to compare between species according to the Bartlett test results.

The principal component analysis (PCoA) and non-metric multidimensional scaling (NMDS) were conducted to test the differences in the relative abundance of OTUs of gut microbiome between the two species. The analysis of similarity test between the two species was conducted using ANOSIM based on the bray_curtis distance with 999 permutations. The PCoA and NMDS analyses were performed using the package vegan in R (*Oksanen et al., 2013*). Moreover, the linear discriminant analysis effect size (LEfSe) (*Segata et al., 2011*) was conducted to test the differences in the bacterial abundances from phylum to family between the two species. Also, the linear discriminatory analysis (LDA) was conducted to estimate the effect size for each selected classification. In this study, only the bacterial taxa with a log LDA score >4 (over 4 orders of magnitude) were used. LEfSe and LDA analyses were performed using the Galaxy online tools (*Afgan et al., 2018*).

## Gene function prediction

We used PICRUSt to search the GreenGene ID of the corresponding OTU based on the Kyoto Encyclopedia of Genes and Genomes (KEGG) database (*Kanehisa, 2019*) and to predict gene functions. Further, these function genes were classified and assigned to the relevant KEGG pathways (*Langille et al., 2013*). We calculated the numbers of functional genes in each pathway to compare the functional enrichment in gut microbiota between the two species. The Circos diagram was plotted to show the relative abundance and distribution status of KOs genes between the two species. We performed the LEfSe to compare the abundances of function genes from KOs gene-level 1 to level 3 between the two species, thereby determining their differences in gene functions. Moreover, LDA analysis was used to assess the effect size for each three KO gene levels. In this study, only gene functions with a log LDA score >2 were used. Student's $t$-test was used to compare the relative abundance of the KOs gene at different levels. All values were presented as mean $\pm$ SE, and all statistical analyses were conducted at the significance level of $\alpha = 0.05$.
## RESULTS

### Gut microbial profile

We obtained 31,402 raw reads from *C. reevesii* and 31,193 raw reads from *T. s. elegans*. After quality control and filtration, we harvested 27,945 and 24,519 high-quality sequences from *C. reevesii* and *T. s. elegans*, respectively. The average read was $8,745 \pm 399$ (ranging from 7,605–10,751 reads per sample) with a sequence length of $480 \pm 4$ bp (ranging from 326–635 bp) for each sample (Fig. S1). Further, an average of $9,317 \pm 601$ high-quality reads (ranging from 8,308-10,751 reads per sample) and $8,173 \pm 242$ high-quality reads (ranging from 7,605-8,603 reads per sample) were obtained from *C. reevesii* and *T. s. elegans*, respectively. The rarefaction curves based on the Shannon index for all samples showed that sufficient sequence numbers could be obtained for further analysis at the current sequencing depth (Fig. S2). Furthermore, the minimum values of the Good's coverage were more than 99.8%, which meant that the vast majority of gut bacteria could be retrieved from these samples.

We obtained 183 representative sequences of OTUs at the 97% similarity level, and 80 representative sequences after removing the influence of OTUs with a large deviation. Among them, the average OTUs were $29 \pm 1.7$ (ranging from 25–32) and $33 \pm 2.9$ (ranging from 29–40) per sample for *C. reevesii* and *T. s. elegans*, respectively (Table S1). Fifty-four OTUs were shared by the two species, three were unique to *T. s. elegans*, and 23 were unique to *C. reevesii*. These OTUs were clustered into five phyla, seven classes, seven orders, 17 families and 28 genera based on phylogenetic classification for the six fecal samples (Table S1). More specifically, species in the gut microbiota belonged to four families, five classes, six orders, 10 families and 10 genera in *C. reevesii*, and to five phyla, seven classes, seven orders, 17 families and 28 genera in *T. s. elegans* (Table S1).

### Interspecific differences in the gut microbiota

As the unidentified OTUs accounted for up to 15.1% at the family level and 41.6% at the genus, we analyzed the composition and relative abundance of gut bacterial community excluding data at the genus level. Fig. 1 shows the relative abundances of the gut microbiota across taxonomic levels from family to phylum. Considering all six samples as a whole, we found that the fecal microbiota was dominated by species of the phyla Bacteroidetes ($78.02 \pm 8.00\%$), Firmicutes ($20.21 \pm 7.94\%$), Proteobacteria ($1.46 \pm 0.98\%$) and Fusobacteria ($0.31 \pm 0.14\%$), the classes Bacteroidia ($78.02 \pm 8.00\%$), Clostridia ($19.73 \pm 7.85\%$), Gammaproteobacteria ($1.46 \pm 0.98\%$), Erysipelotrichi ($0.48 \pm 0.13\%$) and Fusobacteriia ($0.31 \pm 0.14\%$), the orders Bacteroidales ($78.02 \pm 8.00\%$), Clostridiales ($19.73 \pm 7.85\%$), Erysipelotrichales ($1.87 \pm 1.01\%$), Fusobacteriales ($0.31 \pm 0.14\%$) and Aeromonadales ($0.07 \pm 0.03\%$), and the families Porphyromonadaceae ($43.88 \pm 8.83\%$), Bacteroidaceae ($20.01 \pm 4.85\%$), Lachnospiraceae ($13.89 \pm 5.53\%$), Clostridiaceae ($2.41 \pm 0.91\%$), S24-7 ($1.77 \pm 1.15\%$), Enterobacteriaceae ($1.39 \pm 0.99\%$), Ruminococcaceae ($0.66 \pm 0.09\%$), Erysipelotrichaceae ($0.46 \pm \pm 0.13\%$), Fusobacteriaceae ($0.31 \pm 0.14\%$) and Aeromonadaceae ($0.07 \pm 0.03\%$).

The composition and relative abundance of the gut microbiota at different taxonomic levels showed differences between the two species. For the sake of convenience, the

bacteria with a relative abundance >2% were defined as the dominant taxa. The dominant phyla were Bacteroidetes (63.45 ± 10.38%), Firmicutes (34.12 ± 10.75%) and Proteobacteria (2.40 ±1.80%) in *C. reevesii*, and Bacteroidetes (92.59 ±2.63%) and Firmicutes (6.30 ± 2.77%) in *T. s. elegans* (Fig. 1A). The dominant classes were Bacteroidia (63.45 ± 10.38%), Clostridia (33.60 ± 10.55%) and Gammaproteobacteria (2.40 ± 1.80%) in *C. reevesii*, and Bacteroidia (92.59 ± 2.63%) and Clostridia (5.87 ± 2.69%) in *T. s. elegans* (Fig. 1B). The dominant orders were Bacteroidales (63.45 ± 10.38%), Clostridiales (33.60 ± 10.55%) and Erysipelotrichales (2.82 ± 1.84%) in *C. reevesii*, and Bacteroidales (92.59 ± 2.63%) and Clostridiales (5.87 ± 2.69%) in *T. s. elegans* (Fig. 1C). The dominant families were Bacteroidaceae (30.74 ± 2.95%), Porphyromonadaceae (26.77 ± 10.33%), Lachnospiraceae (3.81 ± 1.92%), Clostridiaceae (3.87 ± 1.27%), S24-7 (3.22 ± 1.98%) and Enterobacteriaceae (2.29 ± 1.84%) in *C. reevesii*, and Porphyromonadaceae (60.99 ± 3.11%), Bacteroidaceae (9.28 ± 2.97%) and Lachnospiraceae (3.81 ± 1.92%) in *T. s. elegans* (Fig. 1D).

Microbes of the phylum Firmicutes, the class Clostridia, the order Clostridiales and the families Bacteroidaceae, Clostridiaceae and Lachnospiraceae were more abundant in *C. reevesii* (Table S3; Fig. 2). Microbes of the phyla Bacteroidetes and Fusobacteria, the classes Bacteroidia and Fusobacteriia, the orders Fusobacteriales and Bacteroidales and the families Porphyromonadaceae and Fusobacteriaceae were more abundant in *T. s. elegans* (Table S3; Fig. 2).

Table S2 shows the alpha-diversity indexes for each sample including the community richness (ACE index), diversity (Shannon diversity), evenness (Shannoneven index), and coverage (the Good's coverage index). Student's $t$ test showed that none of these indexes differed between the two species (all $P > 0.09$). PCoA and NMSD analyses showed no significant differences in bacterial relative abundance between the two species (Fig. 3; anosim: $R = 0.44$, $P = 0.10$).

## The predicted metagenomes

PICRUSt analysis was performed to predict the gene functions in the two turtle species based on the 16S RNA of six fecal samples. These gene functions were predicted into three levels of KEGG functional categories. Among them, metabolism-related genes had an overwhelming proportional advantage, with a relative abundance of up to 45.34 ± 1.38% at the first level (Fig. 4A). Furthermore, the gene functions at the first level also included environmental information processing (17.12 ± 1.43%), genetic information processing (16.21 ± 0.45%), human diseases (15.80 ± 0.43%), cellular processes (4.03 ± 0.33%), organismal systems (0.59 ± 0.06%) and unclassified genes (15.80 ± 0.43%) (Fig. 4A). There were 18 major gene functions at the second level (Fig. 4B), among which, the most abundant gene functions were composed of membrane transport (14.64 ± 1.30%), carbohydrate metabolism (10.96 ± 0.37%), amino acid metabolism (8.80 ± 0.19%), replication and repair (6.94 ± 0.23%), energy metabolism (5.00 ± 0.20%), translation (4.02 ± 0.18%), metabolism of cofactors and vitamins (3.79 ± 0.05%), cell motility (3.37 ± 0.42%), nucleotide metabolism (3.26 ± 0.13%) and transcription (3.06 ± 0.08%) (Fig. 4B). Also, at the third level, the gene functions with the highest relative abundance
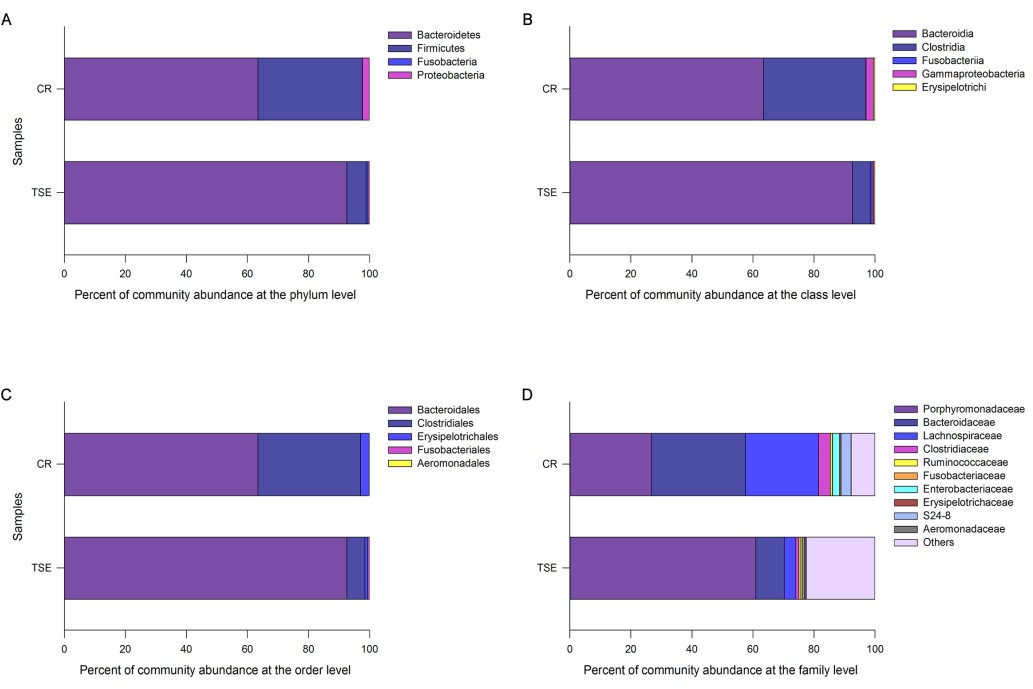

**Figure 1** **Relative abundances of the gut microbiota at the phylum (A), class (B), order (C), and family (D) levels in *Chinemys reevesii* (CR) and *Trachemys scripta elegans* (TSE).** One color indicates one taxon in each plot. The color for others in Plot D indicates all other families not listed in the plot.

were transporters (7.18 ± 0.56%), ABC transporters (3.77 ± 0.40%), DNA repair and recombination proteins (2.25 ± 0.05%), transcription factors (2.21 ± 0.16%) and two-component system (2.19 ± 0.19%) (Fig. 4C and Fig. S3).

As a whole, 240 known KOs and 27 unknown gene functions were identified from the six samples. The distribution of KOs in different species with their relative abundance of >1% was displayed on the Circos diagram (Fig. S3). The gene functions at the first level that accounted for the highest abundance included metabolism (46.76 ± 2.45%), genetic information processing (16.84 ± 0.60%) and environmental information processing (15.48 ± 2.51%) in *C. reevesii*, and metabolism (43.91 ± 0.54%), environmental information processing (18.76 ± 0.30%) and genetic information processing (15.59 ± 0.44%) in *T. s. elegans* (Fig. 4A and Fig. S3). The relative abundance of gene functions at the second level accounted more than 5% were membrane transport (13.13 ± 2.27%), replication and repair (7.32 ± 0.29%), amino acid metabolism (8.93 ±0.31%), carbohydrate metabolism (11.41 ± 0.58%), and energy metabolism (5.26 ± 0.33%) in *C. reevesii*, and those in *T. s. elegans* were membrane transport (16.15 ± 0.16%), replication and repair (6.56 ± 0.18%), amino acid metabolism (8.67 ± 0.17%) and carbohydrate metabolism (10.51 ± 0.27%) (Fig. 4B and Fig. S3). Furthermore, transporters (6.66 ± 0.13% in *C. reevesii*, and 7.69 ± 0.15% in *T. s. elegans*) and ABC transporters (3.28 ± 0.68% in *C. reevesii*, and 4.27 ± 0.15% in *T. s. elegans*) at the third level accounted more than 3% in both species (Fig. 4C and Fig. S3).

## Cladogram

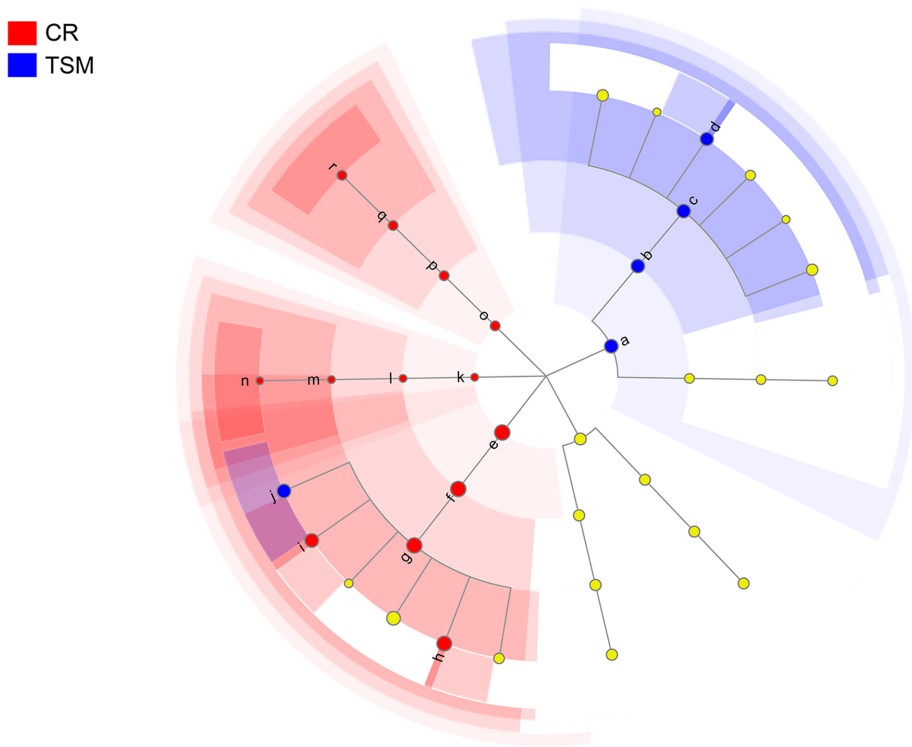

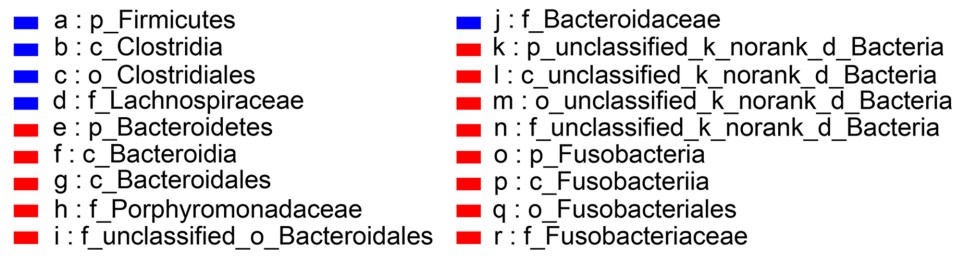

■ a : p_Firmicutes
■ b : c_Clostridia
■ c : o_Clostridiales
■ d : f_Lachnospiraceae
■ e : p_Bacteroidetes
■ f : c_Bacteroidia
■ g : c_Bacteroidales
■ h : f_Porphyromonadaceae
■ i : f_unclassified_o_Bacteroidales

■ j : f_Bacteroidaceae
■ k : p_unclassified_k_norank_d_Bacteria
■ l : c_unclassified_k_norank_d_Bacteria
■ m : o_unclassified_k_norank_d_Bacteria
■ n : f_unclassified_k_norank_d_Bacteria
■ o : p_Fusobacteria
■ p : c_Fusobacteriia
■ q : o_Fusobacteriales
■ r : f_Fusobacteriaceae

**Figure 2** **The differences in relative abundance of gut microbiota between *Chinemys reevesii* and *Trachemys scripta* as shown by linear discriminatory analysis (LDA) scores.** Letters p, c, o and f indicate phylum, class, order and family, respectively. See Fig. 1 for species abbreviations.

LEfSe analysis based on the KOs revealed obvious differences in gene functions between the two species. At the third level, LDA discriminant analysis showed that there was a greater proportion of human disease-related functions (e.g., valine, leucine, and isoleucine biosynthesis, and Influenza A, both $P = 0.046$) and metabolism-related functions (e.g., bisphenol degradation, linoleic acid metabolism, nitrogen metabolism and colorectal cancer, all $P = 0.050$) in *C. reevesii*, as well as tyrosine metabolism ($LDA = 2.79562$, $P = 0.050$) in *T. s. elegans*.

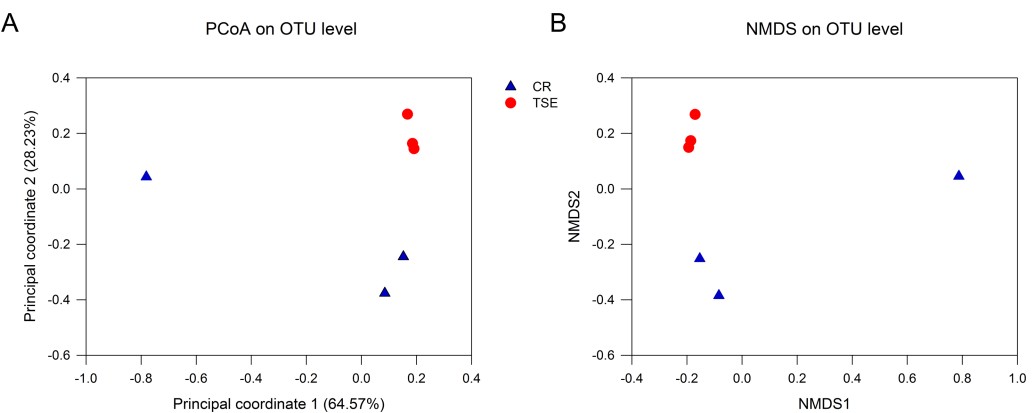

**Figure 3 Fecal microbial diversity in the two turtle species.** Results of principal coordinates analysis (PCoA) and non-metric multidimensional scaling (MDS) of Bray-Curtis distance matrix for bacterial abundance are shown in A and B, respectively. See Fig. 1 for species abbreviations.

## DISCUSSION

From this study where the gut microbiota was compared between two species of turtles raised under the identical conditions for six months we knew the following. First, neither alpha diversity nor beta diversity differed significantly between the two species (Table S2; Fig. 3). Second, the relative abundance of gut microbes differed significantly between the two species across the taxonomic levels from family to phylum (Fig. 2). Third, the relative abundance of metabolism-related functional genes in the gut microbiota differed significantly between the two species, and so did the relative abundance of human disease-related functional genes (Fig. 4).

The influence of the host's genetic background on the gut microbiota has been detected in most vertebrates studied thus far, including fish, reptiles, birds and mammals. For instance, phylogenetic relationships shape the gut microbial profile in phylogenetically closely related mammals that are similar in body shape, craniofacial anatomy and the gut structure (Ley et al., 2008; Kartzinel et al., 2019). The taxonomic or phylogenetic correlates of the gut microbiota has also been reported for birds in Equatorial Guinea (Capunitan et al., 2020). In Galápagos iguanas, differences in the fecal microbial communities are primarily related to the host species and ecotype, and subsequently to population origin (Lankau, Hong & Mackie, 2012). Significant differences in the gut microbiota also exist between the bony fish and sharks (Givens et al., 2015). From these results we know that the influences of phylogeny on the gut microbiota are universal in vertebrates. This is probably the main reason for why we could still distinguish differences in the gut microbiota between two species of turtles even though they had been maintained under identical conditions for six months.

Bacteroidetes and Firmicutes were two dominant phyla in the gut microbiota in both species. This finding is consistent with the studies on other turtle species such as the gopher tortoise *Gopherus polyphemus* (Yuan et al., 2015), the green sea turtle *Chelonia mydas* (Campos et al., 2018; McDermid et al., 2020), and the loggerhead sea turtle *Caretta*

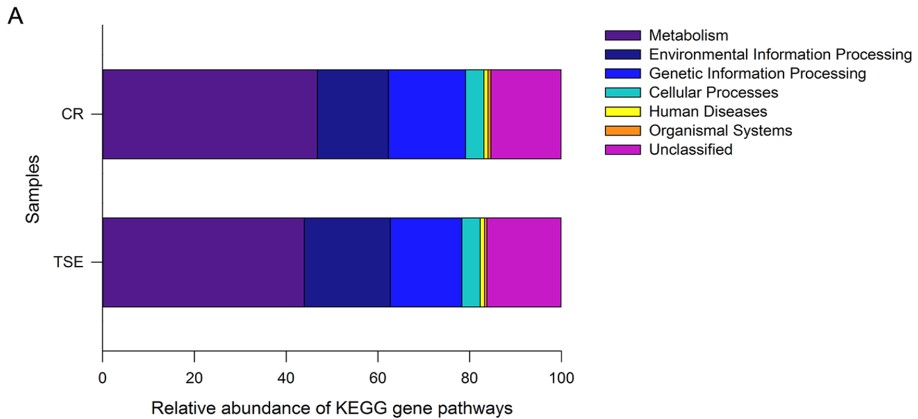

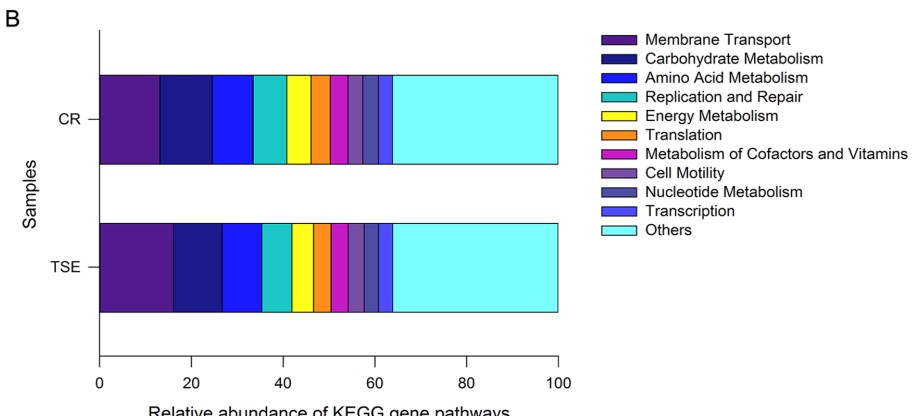

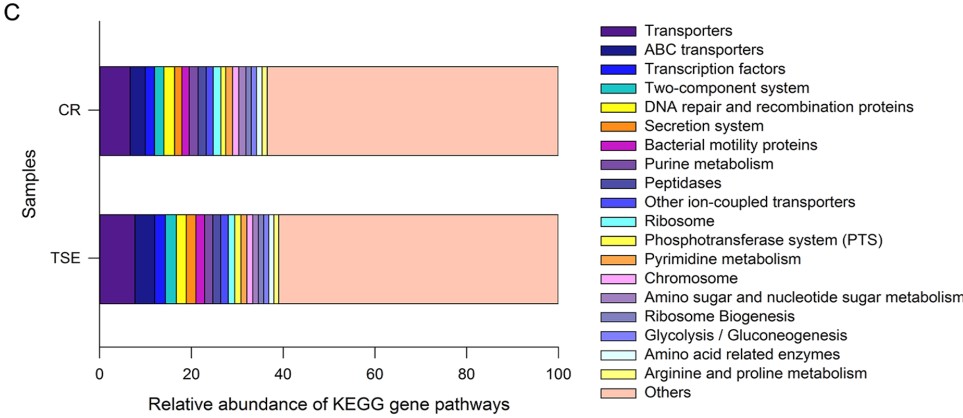

**Figure 4   Gene functional categories based on 16S RNA in the gut microbiota at top (A), second (B) and third (C) levels of relative abundance.** One color indicates one gene function. Detailed descriptions are shown on the right side of each plot. The colors for others in B and C indicate all other gene functions not listed in these two plots. See Fig. 1 for species abbreviations.

*caretta* (*Arizza et al., 2019*), and the painted turtle *Chrysemys picta* (*Fugate, Kapfer & McLaughlin, 2020*). Gut microbes of the families Bacteroidaceae, Clostridiaceae and Lachnospiraceae were comparatively more abundant in *C. reevesii* than in *T. s. elegans* (Fig. 1). Of these three families, Bacteroidaceae belongs to the phylum Bacteroidetes, and Clostridiaceae and Lachnospiraceae both belong to the order Clostridiales, the class Clostridia and the phylum Firmicutes. Furthermore, gut microbes of the families Porphyromonadaceae and Fusobacteriaceae were more abundant in *T. s. elegans* than in *C. reevesii* (Fig. 2). Porphyromonadaceae belongs to the order Bacteroidales, the class Bacteroidia and the phylum Bacteroidetes, and Fusobacteriaceae belongs to the order Fusobacteriales, the class Fusobacteriia and the phylum Fusobacteria (Fig. 2). Microbes of the families Bacteroidaceae and Porphyromonadaceae were the main components of the phylum Bacteroidetes (Fig. 2). Bacteroidetes species in the gut play an important role in degrading carbohydrates and proteins, thus being essential for their hosts in the absorption and utilization of nutrients (*Fernando et al., 2010*; *Nuriel-Ohayon, Neuman & Koren, 2016*). Firmicutes species are universal representatives in mammalian gut microbiota and play important functional roles in digestion and host metabolism (*Ley et al., 2008*). Fusobacteria is usually rare in the gut microbiota of reptiles, but can be obtained from the infected animals and are the dominant phylum in the gut microbiota in scavengers like the American alligator *Alligator mississippiensis* (*Keenan, Engel & Elsey, 2013*), the black vulture *Coragyps atratus* and the turkey vulture *Cathartes aura* (*Mendoza et al., 2018*). The Firmicutes/Bacteroidetes ratio was 0.54 in *C. reevesii* and 0.07 in *T. s. elegans*. It is worth noting that the Firmicutes/Bacteroidetes ratio is negatively correlated with mass gain (*Fernando et al., 2010*). Thus, a lower Firmicutes/Bacteroidetes ratio in *T. s. elegans* suggests a greater potential to gain mass in the species.

Numerous studies on aquatic animals have shown that the gut microbiota is influenced by food ingested, environment, and their age, gender and health status. For example, colony location and captivity influence the gut microbial community composition of the Australian sea lion *Neophoca cinerea* (*Delport et al., 2016*). The gut microbial diversity in green turtles are influenced by their age and population origin (*Campos et al., 2018*). Likewise, the gut microbiota of fish is affected by life stage, tropic level, diet, season, habitat, sex, captivity and phylogeny (*Egerton et al., 2018*). Our experiment design allowed us to exclude the effects of external factors on the gut microbiota because turtles we used were raised under the same conditions for a half year.

Neither alpha diversity nor beta diversity differed between the two species (Table S2, Fig. 3). This might be related to small sample sizes in both species. Data from one *C. reevesii* sample deviated from those from other samples, suggesting that future work could usefully examine the impact of long-term homogeneous culture conditions on the gut microbes in animals. The composition of gut microbes in the two species was relatively simple, presumably because turtles were raised under identical conditions for a long time period (six months). Similar results have been reported in an earlier study of *T. s. elegans* where gut microbes in fecal samples from three adult turtles are mainly composed of Firmicutes and Bacteroidetes (*Du, Zhang & Shi, 2013*). In that study, Bacteroides account for 98.82% of the Bacteroidetes, and Clostridium accounts for 95.81% of the Firmicutes (*Du, Zhang*

*& Shi, 2013*). In another study of *T. s. elegans* the gut microbiota in young turtles is mainly composed of Firmicutes, Bacteroidetes and Proteobacteria (*Peng et al., 2020*).

Our data showed that in both species the most functionally distinct categories were focused on metabolism, genetic information processing and environmental information processing at the first function level, followed by gene functions associated with membrane transport, replication and repair, amino acid metabolism and carbohydrate metabolism at the second level, and transporters and ABC transporters at the third level. In reptiles, that the gene functions of the gut microbes are associated with metabolism has been reported for the timber rattlesnake *Crotalus horridus* (*Mclaughlin, Cochran & Dowd, 2015*), the crocodile lizard *Shinisaurus crocodilurus* (*Tang et al., 2020*), and the northern grass lizard *Takydromus septentrionalis* (*Zhou et al., 2020*). Similar results have also been reported for birds (*Wang et al., 2018*) and mammals (*Zhao et al., 2018*). Previous studies generally support the idea that the most functionally distinct categories of gut microbes play an important role in host energy metabolism.

Comparing gene functions between *C. reevesii* and *T. s. elegans*, we found that they differed in human diseases and metabolic-related functions. Captive animals had an ample opportunity to come into contact with human keepers for transmission of microbiota from host-associated sources, which could colonize the animals. In this study, we found that gut microbes of the families Bacteroidaceae, Clostridiaceae, and Lachnospiraceae were more abundant in *C. reevesii*. Clostridiaceae and Lachnospiraceae (Firmicutes; Clostridia; Clostridiales) are more abundant in the gut microbiota for the time periods of disease progression (*Zhang et al., 2014*). The relative abundance of Bacteroidaceae (Bacteroidetes; Bacteroidia; Bacteroidales) increases at one or two months old in humans and is negatively correlated with depression (*Songjinda et al., 2007*; *Strandwitz et al., 2019*). Gut microbes of the families Porphyromonadaceae and Fusobacteriaceae were comparatively more abundant in *T. s. elegans*. Porphyromonadaceae (Bacteroidetes; Bacteroidia; Bacteroidales) is negatively associated with cognitive decline, affective disorders and anxiety-like behavior in aged mice (*Scott et al., 2017*). Fusobacteriaceae (Fusobacteria; Fusobacteriia; Fusobacteriales) is involved in the fermentative progress of a variety of carbohydrates, amino acids and peptides (*Md Zoqratt et al., 2018*). Therefore, we hypothesized that the difference in gene function between *C. reevesii* and *T. scripta elegans* may be related to their absorption and utilization of food resources. Comparative studies have found that *C. reevesii* is at greater risk of colonizing human gut microbes than *T. s. elegans*, so it is recommended that actions be taken to minimize direct contact between human managers and native turtles.

## CONCLUSION

Our data showed that red-eared slider turtles and Chinese three-keeled pond turtles raised under identical conditions for a long time period (six months) differed in the relative abundance of the microbes and gene functions in the gut microbiota, thus adding evidence for the phylogenetic (genetic) dependence of the gut microbial communities in reptiles. The potential to gain mass was greater in *T. s. elegans* than in *C. reevesii*, as revealed by the

fact that the Firmicutes/Bacteroidetes ratio was lower in the former species. The percentage of human disease-related functional genes was lower in *T. s. elegans* than in *C. reevesii*, presumably suggesting an enhanced potential to colonize new habitats in the former species. Taken together, our data allow the conclusion that the invasive red-eared slider turtle is more successful than the native Chinese three-keeled pond turtle.

## ACKNOWLEDGEMENTS

We would like to thank Zhe Liu and Rong-Fang Wang for help with feeding turtles.

### Funding

The work was supported by grants from the Natural Science Foundation of Jiangsu Province (BK20161556), the Priority Academic Program Development of Jiangsu Higher Education Institutions (PAPD) of Jiangsu Higher Education Institutions and the Finance Science and Technology Project of Hainan Province (ZDKJ2016009-1-2). The funders had no role in study design, data collection and analysis, decision to publish, or preparation of the manuscript.

### Grant Disclosures

The following grant information was disclosed by the authors:
Natural Science Foundation of Jiangsu Province: BK20161556.
Priority Academic Program Development of Jiangsu Higher Education Institutions (PAPD) of Jiangsu Higher Education Institutions.
Finance Science and Technology Project of Hainan Province: ZDKJ2016009-1-2.

### Competing Interests

The authors declare there are no competing interests.

### Author Contributions

- Yan-Fu Qu conceived and designed the experiments, performed the experiments, analyzed the data, prepared figures and/or tables, authored or reviewed drafts of the paper, and approved the final draft.
- Yan-Qing Wu, Yu Du and Peng Li performed the experiments, authored or reviewed drafts of the paper, and approved the final draft.
- Yu-Tian Zhao performed the experiments, analyzed the data, prepared figures and/or tables, authored or reviewed drafts of the paper, and approved the final draft.
- Long-Hui Lin conceived and designed the experiments, authored or reviewed drafts of the paper, and approved the final draft.
- Hong Li analyzed the data, authored or reviewed drafts of the paper, and approved the final draft.
- Xiang Ji conceived and designed the experiments, analyzed the data, authored or reviewed drafts of the paper, and approved the final draft.

## Animal Ethics

The following information was supplied relating to ethical approvals (i.e., approving body and any reference numbers):

This study was performed in accordance with the current laws on animal welfare and research in China, and was approved by the Animal Research Ethics Committee of Nanjing Normal University (IACUC-200422).

## Data Availability

Data are available at the National Center for Biotechnology Information (NCBI) Bioproject database: PRJNA645767.

## Supplemental Information

Supplemental information for this article can be found online at http://dx.doi.org/10.7717/peerj.10271#supplemental-information.

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
