# Peer review of "The invasive red-eared slider turtle is more successful than the native Chinese three-keeled pond turtle: evidence from the gut microbiota"

_PeerJ, doi:10.7717/peerj.10271_

## Round 0.1 · original submission · Major Revisions

I obtained comments from three experts in this field and they all found merit in this study. However, they also raised critical comments to improve the quality and clarity. I will be happy to consider a revised manuscript upon these comments being carefully considered.

Reviewer 1 ·

Basic reporting

This is a very interesting study which I feel should be published. There are very few studies which examine the microbiota of reptiles, such as turtles. However, I do have some questions about your Experimental Design and Validity of Findings.

Experimental design

According to the Materials and Methods one fecal sample/turtle was collected after 6 months in your laboratory. Why collect only one sample/turtle at the 6 months’ period of time? Fecal samples could have been collected starting at time =0, 1 month, 2 months, 3 months, 4 months, 5 months and 6 months. You could have determined how the bacterial diversity changed over time and if the gut microbiota determined in Figure 1 is stable.

Validity of the findings

Unfortunately, I do not think it is valid of say (Line 48 and 49) “The invasive species is more successful than the native species, as revealed by two lines of evidence.” In your study you only have 3 samples/ turtle species and one time point (6 months). I think you need to say “The invasive species could be more successful than the native species as documented in this preliminary study.” Parts of your manuscript, beginning with the title of the manuscript, should be rewritten to reflect this change.
Follow up studies are going to be needed, including fecal samples from both turtle species living in the wild, to make a more definitive conclusion. I see this current study as a first step.

Additional comments

1. Under Keywords add “turtle” since this is a study about turtles.
2. In the Introduction there is nothing mentioned about turtles. I would begin by describing the Chinese three-keeled pond turtle and the red-eared slider turtle. You state the red-eared slider turtle is invasive. Where did the turtle originally come from? Is it causing harm?
3. Line 110 “Roach 454 platform” should be changed to “roche 454 platform”
4. Line 131 Sample Sequencing. Was there any library preparation/adapters used? This needs to be mentioned.
5. Lines 238 and 239 The authors write “On the whole, the fecal
239 microbiota was dominated by species of the phyla Bacteroidetes (78.02±8.00%)…” Are you averaging the abundance of Bacteroidetes from all six samples to get 78.02%?
6. Line 294 The authors wrote “In the whole” This should be changed to “As a whole”
7. Lines 341 Since there are relatively few turtle species which have been studied you should discuss the results of the study by Fugate et al. Curr Microbiol 2020 on the painted turtle.
8. Line 367 The study by McLaughlin et al. Mol Biol Rep (2015) examined the gut microbiota of a Timber rattlesnake using a metagenomics approach. Functional categories are discussed in their paper.
9. Line 711 The correct citation is Md Zoqratt MZH, Eng WWH, Thai BT, Austin CM, Gan HM. Microbiome analysis of Pacific white shrimp gut and rearing water from Malaysia and Vietnam: implications for aquaculture research and management.

Reviewer 2 ·

Basic reporting

The authors assessed the gut microbiome of native Chinese three-keeled pond turtle and the invasive red-eared slider turtle by 16S rRNA sequencing approach, and discovered that the gut microbial diversity and richness of the gut microbiomes were different in two species.

Experimental design

The most important for this work is that the sample size is too small

Validity of the findings

No comments

Additional comments

The authors assessed the gut microbiome of native Chinese three-keeled pond turtle and the invasive red-eared slider turtle by 16S rRNA sequencing approach, and discovered that the gut microbial diversity and richness of the gut microbiomes were different in two species. I have some concerns as listed below. I suggest the authors to carefully revise their manuscript before further process.
Major concerns:
1. The most important for this work is that the sample size is too small, and Especially, In figure 3, there is individual in CR is different with the other two in the same group, which can affect the results.
2. In the abstract, the authors said that the percentage of human disease-related functional genes was lower in T. s. elegans than in C. reevesii, suggesting a greater potential to colonize new habitats in the former species. Are you sure you can get this conclusion? as we know, turtles have lower defence mechanisms to protect them from disease compared with Human, and why compared with human?
3. Gut microbiology may indicate species differences in diet, physiology, and other internal and external factors, and significantly more variation in faecal microbiome composition may enable them to adapt quicker to new environments, so I suggest that the authors compare and find the main OTU or bacteria species in turtles, which is helpful for turtles conservation and farm.


Referece:

Shanmuganandam S, Hu Y, Strive T, Schwessinger B, Hall RN. 2020. Uncovering the microbiome of invasive sympatric European brown hares and European rabbits in Australia. PeerJ 8:e9564 https://doi.org/10.7717/peerj.9564

Annotated reviews are not available for download in order to protect the identity of reviewers who chose to remain anonymous.

Reviewer 3 ·

Basic reporting

This manuscript studied the invasive red-eared slider turtle is more
successful than the native Chinese three-keeled pond
turtle from the gut microbiota angle. It is clear and interesting.

Experimental design

The experimental design is appropriate. The data supports their conclusions.

Validity of the findings

The findings is effective. The invasive species harbors more microbes associated with gain mass. In addition, the the percentage of human disease-related functional genes was less in the invasive species.

Additional comments

This study mainly compared the microbes and gene functions in the invasive and native turtle. They found that the invasive animal harbors more microbes associated with gain mass. In addition, the the percentage of human disease-related functional genes was less in the invasive species. The experimental procedures are very detailed. And the experimental design is reasonable. In addition, the subjects of research are novel. However, there are still some problems in this paper.
1. The distance of one letter in front of each paragraph rather than two Chinese characters
2. Whether to record the altitude, longitude and latitude of the sampling site and the temperature at the time of sampling
3. Why are the LDA scores distinct in difference of gene function and difference of bacterial abundances
4. Please give a clear definition of the ‘dominant microbes’ in this study.
5. It is suggested to supplement the composition of gut microbiota at genus level.
6. How to define ‘more successful’ in this study.
7. What statistical software and drawing software were used in this study?
8. The abbreviation, such as T. s. elegans, should be stated in the previous article firstly, and can‘t be directly exported.
9. Latin names should be marked When a noun appears firstly.
10. From Line 294 to Line 316, the description of the results was not clear enough. It is suggested to reorganize the language.

---

## Round 0.2 · accepted · Accept

I can see you revised the manuscript according to the reviewers' comments and thus am happy to accept your manuscript for publication. Thank you very much for submitting the interesting manuscript to PeerJ.

Reviewer 1 ·

Basic reporting

no comment

Experimental design

no comment

Validity of the findings

no comment

Additional comments

This is an interesting study which should be published.